# Long Term Immune Response Produced by the SputnikV Vaccine

**DOI:** 10.3390/ijms222011211

**Published:** 2021-10-18

**Authors:** Ekaterina Martynova, Shaimaa Hamza, Ekaterina E. Garanina, Emmanuel Kabwe, Maria Markelova, Venera Shakirova, Ilsiyar M. Khaertynova, Neha Kaushal, Manoj Baranwal, Albert A. Rizvanov, Richard A. Urbanowicz, Svetlana F. Khaiboullina

**Affiliations:** 1IFMIB, Kazan Federal University, 420008 Kazan, Russia; ignietferro.venivedivici@gmail.com (E.M.); shaimaa.hamza@mail.ru (S.H.); kathryn.cherenkova@gmail.com (E.E.G.); emmanuelkabwe@ymail.com (E.K.); mimarkelova@gmail.com (M.M.); rizvanov@gmail.com (A.A.R.); 2Kazan State Medical Academy, 420012 Kazan, Russia; vene-shakirova@yandex.ru; 3Department of Infectious Diseases, Kazan State Medical Academy, 420012 Kazan, Russia; I.khaertynova@gmail.com; 4Department of Biotechnology, Thapar Institute of Engineering and Technology, Patiala 147004, India; kneha0563.nk@gmail.com (N.K.); manoj.baranwal@thapar.edu (M.B.); 5Department of Infection Biology and Microbiomes, Institute of Infection, Veterinary and Ecological Sciences, University of Liverpool, Liverpool L3 5RF, UK; Richard.Urbanowicz@liverpool.ac.uk

**Keywords:** SARS-CoV-2, SputnikV, Spike protein, humoral immune response, cellular immunity

## Abstract

SputnikV is a vaccine against SARS-CoV-2 developed by the Gamaleya National Research Centre for Epidemiology and Microbiology. The vaccine has been shown to induce both humoral and cellular immune responses, yet the mechanisms remain largely unknown. Forty SputnikV vaccinated individuals were included in this study which aimed to demonstrate the location of immunogenic domains of the SARS-CoV-2 S protein using an overlapping peptide library. Additionally, cytokines in the serum of vaccinated and convalescent COVID-19 patients were analyzed. We have found antibodies from both vaccinated and convalescent sera bind to immunogenic regions located in multiple domains of SARS-CoV-2 S protein, including Receptor Binding Domain (RBD), N-terminal Domain (NTD), Fusion Protein (FP) and Heptad Repeats (HRs). Interestingly, many peptides were recognized by immunized and convalescent serum antibodies and correspond to conserved regions in circulating variants of SARS-CoV-2. This breadth of reactivity was still evident 90 days after the first dose of the vaccine, showing that the vaccine has induced a prolonged response. As evidenced by the activation of T cells, cellular immunity strongly suggests the high potency of the SputnikV vaccine against SARS-CoV-2 infection.

## 1. Introduction

The outbreak of acute pneumonia in Wuhan, China, in December 2019, spread rapidly throughout the world, prompting WHO to declare a pandemic by 11 March 2020 [1]. Sequence analysis identified a novel coronavirus as the cause of infection [2], which was later named severe acute respiratory syndrome coronavirus 2 (SARS-CoV-2) on 11 February 2020, by the International Committee on Taxonomy of Viruses (ICTV) [1]. Soon after, WHO officially named the disease coronavirus disease 2019 (COVID-19) [1]. The disease rapidly spread from China, leading to a global health emergency. The disease quickly spread, and there are more than 213 million cases documented worldwide, with 4,459,381 deaths confirmed as of 27 August 2021 [3]. COVID-19 has been present in Russia since March 2020, where initially, all confirmed cases were imported, but local transmission was quickly observed and documented [4].

Initially, multiple approaches were suggested to control the disease using repurposed and novel drugs [5,6,7,8]. Some of the repurposed drugs had demonstrated efficacy in vitro [9], yet they failed to translate into patient treatment due to high mortality [10]. Lopinavir was used for COVID-19 treatment, although the drug’s efficacy was inconsistent [11,12]. As a result, limited options remained to treat COVID-19, leading to high morbidity and mortality. With the increasing number of cases registered daily and the inefficacy of administrative measures to prevent virus spread, prevention of infection using vaccines appeared to be one of the most effective approaches to dealing with the pandemic. Several different methodologies have been developed including mRNA-based [13,14,15] and adenovirus-based vectors from gorilla [16] and chimpanzee [17,18] adenoviruses.

The SputnikV vaccine developed by Gamaleya National Research Centre for Epidemiology and Microbiology (GRCEM) demonstrating 91.6% efficacy in a clinical trial [19]. This vaccine is recombinant adenovirus (rAd) 26 and rAd5 vector-based, expressing SARS-CoV-2 Spike (S) protein. The selection of this protein is based on its essential role in virus entry as it binds to the receptor and allows for membrane fusion [20,21]. Currently, the vaccine is routinely used to immunize Russian citizens [22], where 22.9% of Russian citizens were fully vaccinated by August 2021 [23,24]. The vaccine is also authorized in over 60 countries worldwide as of 29 July 2021 [25]. Clinical trial studies have demonstrated that the vaccine is safe, and no adverse effects have been demonstrated [19]. There is currently limited knowledge on which immunogenic epitopes of SARS-CoV-2 S protein elicit an antibody response in immunized individuals and how these compare to antibodies produced in a natural infection.

In this study, we aimed to analyze the SputnikV activation of antibody and T cell immune responses. We have identified similarities between an immunization-induced response and that developed after the recovery from SARS-CoV-2 infection. Additionally, we have shown antibody reactivity three months after vaccination, implying the development of memory lymphocytes after immunization. Analysis of the conservation of immunogenic epitopes was also conducted to demonstrate the potential of vaccine efficacy to SARS-CoV-2 variants of concern. Cytokine activation was analyzed to reveal an insight into the mechanisms of immune response activation.

## 2. Results

### 2.1. Anti-SARS-CoV-2 Antibodies in Serum of Convalescent COVID-19 and Vaccinated with Sputnikv

Anti-SARS-CoV-2 antibodies were measured using the Coronapass ELISA test (Figure 1, Appendix A). Samples used for this analysis were collected on days 21 and 42 after the first dose of vaccine and labeled as D21 and D42, respectively. The day of the first vaccine dose was indicated as day 0 and marked as D0. The analysis of D21 samples confirmed that the first dose of SputnikV elicited an antibody response, while the second dose boosted antibody production. These data indicate that SputnikV is immunogenic after a single dose. However, the second dose boosts antibody response significantly higher as compared to the first dose.

Next, we analyzed the humoral immune response in SputnikV immunized and COVID-19 convalescent individuals. We have found that the antibody response in convalescent COVID-19 was higher than that in vaccinated with SputnikV on D21; however, it was lower than that on D42 post-immunization (Figure 1, Appendix A).

### 2.2. Immune Response to SARS-CoV-2 Peptides in Convalescent COVID-19 and Sputnikv Vaccinated Sera

We analyzed the immune response to SARS-CoV-2 S protein peptides in vaccinated and convalescent COVID-19 sera (Figure 2). On D42 after the first dose of the vaccine the sera had higher reactivity to 30 peptides (S9, S10, S11, S15, S16, S17, S18, S21, S23, S24, S25, S27, S28, S35, S37, S39, S42, S45, S46, S47, S48, S56, S59, S65, S66, S68, S69, S70, S72 and S74) as compared to the same individuals before vaccination (D0) (Figure 2A). Those who had recovered from COVID-19 had an increased reactivity to 22 peptides (S3, S9, S10, S15, S17, S18, S21, S23, S25, S28, S29, S30, S45, S47, S59, S62, S68, S69, S70, S71, S72 and S74) when compared to control sera collected before vaccination (Figure 2B). 

Evaluating each panel of reactive peptides, we identified two highlights. First, there were 17 peptides (S9, S10, S15, S17, S18, S21, S23, S25, S28, S45, S47, S59, S68, S69, S70, S72 and S74) recognized in both immunized and convalescent individuals (77.3% and 56.6% of all reactive peptides in vaccinated and convalescent samples, respectively). These data suggest that many peptides identified in COVID-19 convalescent sera are also recognized in the sera from SputnikV vaccinated individuals. Second, there were also peptides, uniquely recognized in immunized sera (S11, S16, S24, S27, S35, S37, S39, S42, S46, S48, S56, S65 and S66) and in convalescent COVID-19 sera (S3, S29, S30, S62 and S71). These data indicate induction of a distinct immune response after SARS-CoV-2 vaccination and infection.

We also found that reactivity to S3, S15, S23, S28, S29, S30, S45, S62, S68, S70 and S71 was higher in COVID-19 convalescent compared to vaccinated sera (Figure 2(C-I)). In contrast, reactivity to S11, S16, S18, S21, S24, S27, S35, S37, S39, S42, S48, S56, S65 and S66 was higher in vaccinated as compared to COVID-19 convalescent sera (Figure 2(C-II)).

PCA analysis demonstrated differences in the reactivity of the antibody response in SARS-CoV-2 infected and fully vaccinated individuals (Figure 3). Before immunization, the reactivity of serum samples with SARS-CoV-2 S peptides forms a tight cluster suggesting that the immune status of volunteers was similar. After vaccination, the reactivity to SARS-CoV-2 S peptides changes and the cluster’s shape altered, yet it still formed a distinct group. In contrast, COVID-19 convalescent sera reactivity to SARS-CoV-2 S peptides formed a larger spread cluster, suggesting that the immune response had more variability than that in vaccinated. The PCA analysis also demonstrated that immunized and convalescent COVID-19 sets overlap, indicating similarity in the immune reactivity between these two groups. This data demonstrates that all volunteers had no history of SARS-CoV-2 exposure, supporting their initial statement of having no COVID-19 infection.

### 2.3. Mapping of Reactive Peptides on S Protein

Next, we analyzed the location of these peptides on a map of the S protein. We have found that peptides recognized by immunized and convalescent sera are located in several domains of the S protein: N terminal Domain (NTD) of the receptor-binding domain (RBD), as well as in the fusion peptide (FP) and heptad repeat (HR) functional regions (Figure 4). When peptides identified in immunized individuals were mapped onto the S protein active regions, they were found to locate in the S1 (NTD and RBD) and S2 (FP, HR1 and HR2) domains (Figure 4A). Similarly, peptides that were recognized in COVID-19 convalescent sera were also located in S1 (NTD and RBD domains) as well as in S2 (FP and HR) (Figure 4B). These data demonstrate that the SputnikV induced immune response targets all significant domains in the S protein, suggesting that vaccine-induced antibodies could be effective at neutralizing entry of the virus.

Seventeen peptides that were recognized in both vaccinated and COVID-19 convalescent sera were mapped on the three-dimensional model of S protein (Figure 5A; yellow). Eight of these peptides were located on the S1 subunit, from which five (S9, S10, S15, S17 and S18) and three (S23, S25 and S28) peptides were present in NTD and RBD domains, respectively (Figure 5A,B). Nine peptides were present in the S2 and TM domains, where FP and HR2 contain one (S47) and two (S69 and S70) peptides recognized in both immunized and convalescent sera, respectively.

Thirteen peptides were recognized exclusively with Sputnik vaccinated sera, in which eight (S11, S16, S21, S24, S27, S35, S37 and S38) and five (S42, S48, S56, S65 and S66) peptides were in S1 and S2 subunit, respectively (Figure 5C,D; red). This group had two (S11 and S16) and three (S21, S24 and S27) peptides in the NTD and RBD domains, while the FP and HR1 regions had one peptide each. Six peptides were recognized exclusively by convalescent COVID-19 sera, where S3, S19, S29 and S38 were located in S1, while S62 and S71 were in S2 protein (Figure 5E,F; green).

### 2.4. Peptide Conservation Analysis 

The prolonged efficacy of a vaccine-induced immune response is the primary goal of immunization. This will, in part, depend on the stability of the immunogenic epitopes in the virus. If the changes are too significant, then antibodies will fail to recognize the epitopes, which could reduce protection [26,27]. To address this important issue, we conducted a conservation analysis of the viral peptides recognized in SputnikV immunized individuals and convalescent COVID-19 sera in currently circulating strains. We included 52 SARS-CoV-2 virus strains in the analysis identified from around the world (Appendix A). The sequences of 36 peptides, recognized by the serum from immunized individuals, were analyzed against all SARS-CoV-2 strains selected (Table 1). We found that 12 peptides (S16, S17, S18, S19, S35, S47, S48, S59, S62, S65, S68 and S72) were 100% identical compared to all virus strains included in this study. These peptides are located in both the S1 and S2 domains of the SARS-CoV-2 virus. We also found 15 more peptides with various degrees of conservation (90.4–98.1%) to the selected virus strains. Together, 27 out of 36 peptides (75%) had preservation of 90.4–100% to the viral strains, suggestive of a high probability of interaction between immunogenic virus epitopes and antibodies.

Next, we divided all peptides into six groups based on the proportion of conservation to the SARS-CoV-2 variants: (1) 100% conservation; (2) >90 < 100% conservation; (3) >80 < 90% conservation; (4) >70 < 80% conservation; (5) >60 < 70% conservation and (6) <60% conservation. The complete 20-mer peptide sequences were stratified depending on whether sera from vaccinated, convalescent, or both recognized them (Table 2). We found that out of 12 peptides having 100% conservation in 52 strains included in the analysis, six (50%) were recognized by both vaccinated and convalescent sera. Four were recognized by vaccinated sera and two by convalescent sera. A similar pattern was observed in the group of peptides with 90–100% amino acid (aa) similarity, with more peptides recognized by vaccinated compared to convalescent sera.

### 2.5. Mutational Analysis of Reactive Peptides

Mutational analysis was carried out using 52 SARS-CoV-2 virus strains (Appendix A) against the reference sequence (YP_009724390.1) used to generate the peptide library. Eighty-seven positions were found to have a potential mutation with a frequency less than 0.1 in 76 positions (Appendix A). All peptides recognized by vaccinated and convalescent COVID-19 sera were analyzed for the presence of these mutations. Out of 36, 12 peptides were found to have none of these mutations. Of the remaining 24 peptides identified as having the mutation, six peptides (S21, S45, S56, S69, S70 and S71) had a single mutation (Table 3). Additionally, 13 peptides (S3, S11, S15, S23, S24, S25, S27, S28, S30, S42, S46, S66 and S74) had two mutations each. Only five peptides had a substantial, up to 7, number of mutations, which could affect the structure of the peptide and its immunogenicity. We also mapped the mutations found in B.1.617.2 (Delta variant; Accession no QVI56963), a fast-spreading variant of SARS-CoV-2 [28], on S protein peptides recognized by vaccine serum. Five mutations were found in four S protein peptides: S9 (aa 142 G > D, aa 156 E > G), S27 (aa 452 L > R), S28 (aa 478 T > K) and S37 (aa 614 D > G). These four peptides represent 11.1%, a small fraction of all peptides reactive to SputnikV immunized sera. Therefore, it could be suggested that the humoral immune response induced by immunization will be effective against the delta variant.

### 2.6. T Cell Immune Response to SputnikV Vaccine

The vaccine’s protective efficacy is also determined by activation of the T cell immune response, which contributes to the clearance of infected cells [29]. The T cell component of the anti-SARS-CoV-2 immune response can be analyzed by the release of IFN-γ by T leukocytes upon exposure to antigens [30]. Therefore, we sought to demonstrate T cell activation by SputnikV using a Tigra test. An increased number of IFN-γ spots were detected in a Tigra test from 26 immunized individuals at day 90 (D90) compared to 12 anti-SARS-CoV-2 antibody-negative controls (Figure 6A). Our data provide strong evidence for the ability of SputnikV to induce a T cell immune response. We also observed anti-SARS-CoV-2 antibodies in samples taken on day 90 (D90) after the first dose of vaccine (Figure 6B), implying successful generation of both humoral and cellular immunological memory.

To demonstrate that SputnikV can induce a long-term immune response, blood and serum samples from the same 26 immunized individuals were collected on day 210 (D210) after the first dose of vaccine and used to analyze T cell reactivity and presence of antibodies using a Tigra test and a Coronapass test, respectively (Figure 6A,B). We have found that T cell reactivity and antibodies remained significantly increased 210 days after immunization.

### 2.7. Cytokine Analysis

Cytokines are biologically active molecules essential for the induction, maintenance and long-term support of the immune response [31,32]. These molecules are produced upon exposure to an antigen, which can be introduced via natural (infection) or artificial (vaccination) exposure [31,33,34,35]. To determine whether cytokine activation in COVID-19 convalescent and SputnikV immunized individuals is similar, we analyzed the serum level of 48 cytokines. The PCA analysis revealed that changes in serum cytokines are similar in vaccinated individuals at D21 and D42, forming a single, well-defined cluster (Figure 7). In contrast, more variations in cytokine levels were identified in COVID-19 serum 42.0 ± 11.1 days after infection. These data suggest that cytokine activation due to SputnikV vaccination is more uniform as compared to that after infection.

Next, we analyzed changes in serum cytokine levels in SputnikV vaccinated individuals at 21 (D21) and 42 (D42) days after the first dose (D0) of the vaccine (Figure 8A,B). We have found that 14 cytokines were increased (IL-1α, IL-2Ra, IL-3, IL-10, IL-12p70, IL-13, CCL7, IFN-α2, bFGF, GM-CSF, LIF, M-CSF, b-NGF and TRAIL), while 25 cytokines were decreased (IL-1Ra, IL-2, IL-4, IL-5, IL-7, IL-8, IL-9, IL-12p40, IL-16, IL-17, CCL2, CCL3, CCL4, CCL5, CCL11, CXCL9, CXCL10, CXCL12, IFN-γ, HGF, MIF, SCF, SCGF-b, TNF-α and TNF-β) in serum 21 days after immunization as compared to D0. The pattern of activated cytokines remained mainly similar 42 days after vaccination, where the level of 12 cytokines were higher (IL-1α, IL-2Ra, IL-10, IL-12p70, IL-13, CCL7, IFN-α2, GM-CSF, LIF, M-CSF, b-NGF and TRAIL) and 26 cytokines were lower (IL-1Ra, IL-2, IL-4, IL-5, IL-6, IL-7, IL-8, IL-9, IL-12p40, IL-16, IL-17, CCL2, CCL3, CCL4, CCL5, CCL11, CXCL9, CXCL10, CXCL12, IFN-γ, HGF, MIF, SCF, SCGF-b, TNF-α, TNF-β) as compared to serum before vaccination. Overall, 12 cytokines (IL-1α, IL-2Ra, IL-10, IL-12p70, IL-13, CCL7, IFN-α2, GM-CSF, LIF, M-CSF, b-NGF and TRAIL) remained activated after first (D21) and second (D42) doses of vaccine. Twenty-five cytokines (IL-1Ra, IL-2, IL-4, IL-5, IL-7, IL-8, IL-9, IL-12p40, IL-16, IL-17, CCL2, CCL3, CCL4, CCL5, CCL11, CXCL9, CXCL10, CXCL12, IFN-γ, HGF, MIF, SCF, SCGF-b, TNF-α and TNF-β) were consistently lower in immunized individuals as compared to that before the vaccination. Interestingly, serum levels of IL-6, a cytokine identified as playing an essential role in the pathogenesis of COVID-19 [36], were decreased after the second dose of vaccine.

Then we sought to analyze cytokine activation in SputnikV immunized and convalescent SARS-CoV-2 sera. We identified three groups: cytokines affected in (A) both vaccinated and COVID-19, (B) vaccinated only, and (C) COVID-19 only (Figure 9). We have found that IL-1α, IL-3, IL-10, IL-12p70, CCL7, IFN-α2, bFGF, LIF and TRAIL were upregulated in both vaccinated and convalescent COVID-19 sera (Figure 9(A-I)). Additionally, CCL3, CCL11, HGF and IL-6 were lower in both vaccinated and convalescent COVID-19 sera (Figure 9(A-II)). Next, we identified a group of cytokines activated after COVID-19, such as IL-2, IL-4, IL-5, IL-12p40, IL-17 and MIF, while being lower in vaccinated than before vaccination (Figure 9(A-III)). In contrast, only one cytokine, b-NGF, was lower in convalescent COVID-19 and increased in vaccinated sera (Figure 9(A-IV)).

Next, we identified cytokines affected only in vaccinated individuals. We found multiple downregulated cytokines (IL-1Ra, IL-7, IL-8, IL-9, IL-16, CCL2, CCL4, CCL5, CXCL9, CXCL10, CXCL12, IFN-γ, SCF, SCGF-b, TNF-α and TFN-β) (Figure 9(B-I)). At the same time, IL-2Ra, IL-13, GM-CSF and M-CSF were upregulated in SputnikV vaccinated individuals (Figure 9(B-II)). There was also a group of cytokines activated only in COVID-19 sera (Figure 9C), while their level remained unaffected in vaccinated individuals compared to before vaccination.

These data strongly indicate that, although multiple cytokines were affected similarly in vaccinated and convalescent COVID-19 sera, there were some differences in cytokine activation, which could account for variation in immune response and clinical symptoms.

## 3. Discussion

Our data confirm the immunogenicity of the SputnikV vaccine, developed by GRCEM [19], where humoral and T cell immune responses were detected in vaccinated individuals. Our data corroborate the previous report by Logunov et al., where anti-SARS-CoV-2 antibodies and activated T cells were detected in immunized individuals [37]. In our study, we further advanced the understanding of the SputnikV vaccine by demonstrating the S protein epitopes involved in the induction of an immune response. We have identified several immunogenic epitopes located in NTD, RBD, FP, HR1 and HR2. Sixteen peptides were found in S1, and fourteen peptides were in S2 domains. S1 and S2 contain several domains and epitopes, ranging from binding to the receptor, membrane fusion and entry [20]. Therefore, by targeting multiple regions, antibodies in immunized individuals can potentially interfere with the most important events in virus replication: entry to the target cell.

Interestingly, six peptides reacting with immunized serum were in the RBD of S1 responsible for binding to the ACE2 receptor [38]. It is the primary target for the most potent neutralizing antibodies [39,40]. Therefore, it is expected that antibodies induced by SputnikV could also be neutralizing, which was previously shown by Logunov et al. [37]. Additionally, RBD targeting vaccines were shown to be highly immunogenic [41,42] in part due to eliciting neutralizing antibodies against SARS-CoV. We also identified peptides in NTD, which were shown to contain epitopes producing neutralizing antibodies [43]. These antibodies appear to interfere with receptor binding or S protein conformation [44]. Our data suggest that neutralizing antibodies, previously detected in immunized individuals, could target RBD and NTD. Together, antibodies to RBD and NTD could provide broad coverage of epitopes in the RBD and FP of SARS-CoV-2, which will reduce the virus’s potential to escape from a vaccine-induced immune response.

We also identified multiple peptides in the S2 domain located in FP, HR1 and HR2. Upon attachment to the receptor, FP becomes embedded into the target cell membrane and installs an anchor inside to initiate fusion with the viral envelope [45,46]. HR1 and HR2 also contribute to fusion by bringing viral and cellular membranes proximal to each other [47]. Therefore, antibodies targeting these regions could potentially interfere with membrane fusion. As HR1 and HR2 are highly conserved among SARS-CoV viruses [48,49], including SARS-CoV-2, antibodies to these regions of S protein could prevent virus escape from immune response induced by the vaccine.

Convalescent plasma was shown to effectively treat critically ill COVID-19 patients [50,51]. This treatment is a polyclonal mix of antibodies developed to many viral proteins, including the S protein. It appears that among all antibodies, those recognizing the S protein are the most important in preventing a severe form of COVID-19. This assumption is based on the observation of Dispinseri et al., demonstrating that compromised immune responses to the S protein were major traits of critical COVID-19 conditions [52]. We have identified multiple peptides recognized by sera from both immunized and convalescent COVID-19 individuals. These data suggest that the SputnikV vaccine induces a humoral immune response similar to that occurring naturally after infection. Therefore, our data could explain the protective efficacy of the SputnikV vaccine demonstrated by Logunov et al. [37]. We believe that selection of epitopes similar to those in naturally infected and recovered individuals is the mechanism of SputnikV protection against COVID-19.

Studies have demonstrated that a single aa replacement could help the virus escape elimination by neutralizing antibodies [53,54]. Concerns have been presented that some mutations increase the infectivity of SARS-CoV-2 [55,56] and that new variants of SARS-CoV-2 may mutate to escape vaccine-induced immunity [57]. Our analysis demonstrated that several peptides recognized in vaccinated individuals remained 100% conserved in SARS-CoV-2 strains worldwide. Not surprisingly, three peptides were located in NTD, a highly conserved region [58,59]. We also identified four peptides in the RBD, which were 90–100% conserved in all SARS-CoV-2 strains included in the analysis. RBD region mutations could affect the neutralizing capacity of antibodies in the later waves of pandemics [21]. Our finding of conserved peptides in NTD and FP provides evidence that SputnikV induced antibodies should retain antiviral efficacy even in the case of a mutation in the S protein. We also did a detailed analysis of S protein mutations in one of the fastest spreading SARS-CoV-2 viruses, B.1.617.2 (Delta variant), which is currently detected in 78 countries [3,60]. This variant has shown moderate resistance to vaccine-induced immunity [28]. Only four peptides recognized by the immunized serum contained the aa positions mutated in the Delta variant of SARS-CoV-2. This is 11.1% out of all reacting peptides and represents a small fraction. Therefore, we suggested that the humoral immune response induced by SputnikV will be effective against the Delta variant of SARS-CoV-2.

We have identified multiple regions on S protein-containing immunogenic epitopes. These regions could be grouped as aa 137–309, 341–471, 579–812 and 1084–1188. Interestingly, these regions are within those Grifoni A et al. predicted as associated with robust immune response [61]. A similar location of the peptides reacting to COVID-19 sera was demonstrated by Shrock et al. [62]. Many peptides that were recognized by convalescent COVID-19 sera were also recognized by vaccinated sera. These data provide strong evidence that the immune response induced by the vaccine is similar to that of a naturally developed response, after recovery from SARS-CoV-2 infection. Therefore, it could be suggested that the protective efficacies of convalescent and immunized sera are comparable.

Our results confirm that SputnikV induces a T cell immune response supporting findings by Logunov et al. [37]. We have found that the T cell immune response was active 42, 90 and 210 days after immunization, suggesting long term circulation of memory T lymphocytes. These lymphocytes are essential for protection against virus infection [63], providing long-lasting immunity. T cells were shown to be activated in convalescent COVID-19 patients [64]. The analysis also revealed that those activated lymphocytes contribute to the amelioration of clinical symptoms of the disease. Multiple studies have found activation of T cells in convalescent COVID-19, where the S protein appears to maintain this immune response [64,65,66]. Interestingly, several peptides (S10, S21, S23, S46 and S48) that activated T cells in SputnikV immunized individuals were similar to those described by Peng et al. in convalescent COVID-19 patients [64]. The immunogenic epitopes appear in aa 166–180, 351–365, 381–395, 751–765 and 801–815 regions on the S protein. Additionally, we have identified two peptides (S21 and S23) that activated T cell responses in SputnikV vaccinated individuals as located in the RBD of the S protein, suggesting that this domain also contains T cell epitopes similar to that described by Ni et al. in convalescent COVID-19 patients [66]. These data indicate that the T cell response in immunized individuals resembles that of individuals that recovered from a SARS-CoV-2 infection. Our data also show that the SputnikV vaccine-induced T cell and humoral immune responses to the S protein of SARS-CoV-2 lasted out to seven months after vaccination. This suggests the long-term efficacy of this vaccine, which is a significant consideration with all vaccine candidates [67,68].

There is limited real-life data on the duration of the immune response elicited by the Sputnik V vaccine. Our data, therefore, collected on samples from 90 and 210 days after the first dose of SputnikV vaccine demonstrating long-term activation of both the humoral and T cell immune responses is of importance. These long-term immune responses could explain the efficacy of the SputnikV vaccine reported by Gonzalez et al. in an Argentinian cohort [69]. Authors have also shown that receiving even a single dose of vaccine reduced the duration of hospitalization and fatality, after exposure to SARS-CoV-2. In another study by Rossi et al., eliciting anti-SARS-CoV-2 neutralizing antibodies was demonstrated in 288 volunteers 21 days after the first dose vaccine [70]. Additionally, the efficacy of SputnikV was demonstrated in a cohort from Venezuela, where even a single dose was shown sufficient to induce neutralizing antibodies in previously SARS-CoV-2 positive individuals [71]. A 100% seroconversion was found in a Venezuelan cohort 6 weeks after the second dose of SputnikV vaccine, confirming induction of long-term immune response.

Cytokines regulate immune responses during natural infection and vaccination [34,35]. These cytokines also contribute to developing clinical symptoms of COVID-19 and post-immunization side effects [72,73]. We have found fifteen cytokines that remain upregulated in convalescent COVID-19 individuals one month after infection (IL-1α, IL-3, IL-10, IL-12p70, CCL7, IFN-α2, bFGF, LIF, TRAIL, IL-2, IL-4, IL-5, IL-12p40, IL-17 and MIF) indicating a strong activation of the immune response. These cytokines are associated with inflammation, activation of T and B cell immune response, and regeneration [74]. A set of cytokines were upregulated in SputnikV immunized individuals (IL-1α, IL-3, IL-10, IL-12p70, CCL7, IFN-α2, bFGF, LIF and TRAIL), where nine of them were similar to that in convalescent COVID-19. These data suggest similarity in mechanisms of an immune response activation in vaccinated and convalescent individuals.

It should be noted that convalescent COVID-19 sera had more cytokines upregulated 42 days after infection than immunized, indicating that infection generates a more robust immune response. This could be explained by the exposure of the immune system to multiple viral antigens during infection compared to only the S protein used in SputnikV. Additionally, differences in cytokine activation could be explained by the differing quantities of SARS-CoV-2 antigens encountered during infection and upon vaccination. Some findings support this suggestion where a higher viral load was closely related to severe COVID-19 [75,76]. Additionally, the simultaneous exposure to adenovirus antigens from the vector and SARS-CoV-2 antigens could affect cytokine activation in vaccinated individuals. The lesser number of cytokines activated in vaccinated compared to convalescent COVID-19 could also explain the lack of clinical symptoms of COVID-19, such as severe inflammation, high fever and tissue damage [72].

In summary, we have shown that vaccination with SputnikV induces a broad antibody response that recognizes a wide variety of epitopes on the S protein and excellent cellular response. Both of these responses are still measurable three months after vaccination, suggesting the long-term efficacy of this vaccine.

## 4. Materials and Methods

### 4.1. Subjects

Serum samples were collected from 40 individuals vaccinated with SputnikV (23 females and 17 males; mean age 43.3 ± 16.4 years old). Demographic information is summarized in Table 4.

Samples were collected immediately before the first vaccine dose (day 0; (D0)), second vaccine dose (day 21; D21) as well as day 42 (D42) and day 210 (D210) after the first vaccine dose. Schematic representation of vaccination and serum samples collection is presented in Figure 10. An additional 40 serum samples were collected from convalescent COVID-19 patients 42.0 ± 11.1 days post-recovery (25 females and 15 males; mean age 39.1 ± 13.2 years old). Out of 40 SputnikV vaccinated individuals, serum and blood samples were collected from the same 26 vaccinated individuals on each, D90 and D210, after vaccination. Blood samples from 12 controls that were negative for SARS-CoV-2 antibodies and had no symptoms of COVID-19 were also collected. Sample aliquots were stored at -80 °C. Each sample was used only once to avoid freeze-thaw cycles.

Samples were collected before the first dose vaccine day 0 (D0), as well as day 21 (D21) and day 42 (D42) days after the first dose vaccine. Additionally, follow up samples were collected on day 90 (D90) and day 210 (D210) after the first dose of the vaccine.

Red color arrow—days of vaccination; Blue color arrow—day of serum sample collection. Black bar—days of sample collection. 1st—First dose of vaccine; 2nd—Second dose of vaccine.

### 4.2. Inclusion Criteria

Participants had to be older than 18 years, have a negative SARS-CoV-2 PCR result and no IgG, IgM or IgA response in a SARS-CoV-2 ELISA. Before enrolling, the participants completed a COVID-19 questionnaire acknowledging no contact with a COVID-19 infected person in the previous 14 days. No female participants were pregnant at the point of study. No participant had acute or chronic upper respiratory tract infections in the last 14 days.

### 4.3. Ethics Statement

The Kazan Federal University ethics committee approved this study, and signed informed consent was obtained from each patient and controls according to the guidelines adopted under this protocol (protocol 27 of the meeting of the ethics committee of the KFU dated 28 December 2020). Sample collection in 2021 was done according to a protocol approved by the Institutional Review Board of the Kazan Federal University, and informed consent was obtained from each respective subject according to the guidelines approved under this protocol (Article 20, Federal Law “Protection of Health Right of Citizens of Russian Federation” N323-FZ, 21 November 2011).

### 4.4. COVID-19 ELISA

According to the manufacturer’s instructions, the «SARS-CoV-2-CoronaPass» ELISA kit (Genetico, Moscow, Russia) was used to determine SARS-CoV-2 specific antibodies (IgM, IgG and IgA). Briefly, SputnikV vaccinated, and control serum was mixed with conjugate-1 at 1:10 ratio and incubated for 30 min at 37 °C in a 96-well plate with pre-adsorbed SARS-CoV-2 antigens. Inactivated, negative control human serum was provided with the kit. Following washes (3×; 0.5% Tween20 in PBS, PBS-T), wells were incubated with anti-human-IgG + IgM + IgA − HRP conjugated antibodies for 30 min at 37 °C. After incubation and washes (3×; 0.5% Tween20 in PBS), wells were incubated with 3,3′,5,5′ Tetramethylbenzidine (Chema Medica, Moscow, Russia). The reaction was stopped by adding an equal amount of 10% phosphoric acid (TatKhimProduct, Kazan, Russia). Data were measured using a microplate reader (Infinite Pro 200, Tecan, Mannedorf, Switzerland) at OD_450_ with reference OD_650_.

Anti-SARS-CoV-2 antibody-positive results had a value >1.0 when calculated using the equation:

Positive result = OD_450_ tested sample/OD_450_ negative control + 0.15.

### 4.5. COVID-19 Peptide Reactivity

SARS-CoV-2 Spike (S) protein peptides (20 aa) with three aa overlap were synthesized by GeneScript (Jiangsu, China) based on the S protein aa sequence (Gen bank accession number: YP_009724390.1) (Yu, Sun et al. 2020). SARS-CoV-2 S proteins peptide sequences (purity >90%) are summarized in Table 5. The overlapping peptide library was used to analyze reactivity with serum from vaccinated individuals, convalescents and controls. Each peptide was coated at ten µg/mL in 100 µL in duplicate into a 384-well plate and incubated at 4 °C for 18 h. After washing, plates were incubated with serum samples (1:100; 50 µL) at 4 °C for 18 h. Following washes (3×; 0.5% Tween20 in PBS, PBS-T), wells were incubated with anti-human IgG-HRP conjugated antibodies (1:10,000 in PBS-T, American Qualex Technologies, USA) for two h at room temperature. Washed (3×; 0.5% Tween20 in PBS), wells were incubated with 3,3′,5,5′ Tetramethylbenzidine (Chema Medica, Moscow, Russia). The reaction was stopped by adding an equal amount of 10% phosphoric acid (TatKhimProduct, Kazan, Russia). Data were captured using a microplate reader (Infinite Pro 200, Tecan, Switzerland) at OD_450_ with reference OD_650_. 

### 4.6. Peripheral Blood Mononuclear Cells (PBMCs) Isolation

Blood samples were collected in tubes containing 3.2% sodium citrate. PBMCs were isolated using ficoll density gradient (1.077 g/cm^3^, PanEco, Moscow, Russia) according to standard procedures (Hawley et al., 2004 “Flow Cytometry Protocols”) [77]. PBMCs were resuspended in RPMI-1640 medium (PanEco, Moscow, Russia) supplemented with 10% fetal bovine serum (HyClone, city, South Africa), 2 mM L-glutamine (PanEco, Moscow, Russia), 250 U penicillin and 250 μg streptomycin (PanEco, Moscow, Russia).

### 4.7. ELISpot Analysis

T cell reactivity to SARS-CoV-2 S protein was analyzed using a Tigra Test SARS-CoV-2 kit (Generium Corporation, Vladimir region, Russia). S protein antigen (50 µL; provided by the manufacturer) was added into anti-human IFN-γ pre-coated 96-well plates. Then, PBMCs (4 × 10^5^ cells/well) were added in 200 µL of RPMI-1640 medium supplemented with 10% FBS (HyClone, South America), 2 mM L-glutamine (PanEco, Moscow, Russia) and 1% mixture of antibiotics penicillin-streptomycin (PanEco, Moscow, Russia). Cells were incubated for 24 h (37 °C, 5% CO_2_). Then cells were removed, wells washed five times with Dulbecco phosphate-buffered saline (DPBS) and IFN-γ detection antibodies conjugated with alkaline phosphatase were added for one h at room temperature. After washing (4× times, DPBS), wells were incubated with 5-bromo-4-chloro-3-indolyl phosphate/Nitro Blue Tetrazolium (BCIP/NBT) substrate solution for 10 min at room temperature. Membranes were washed, air-dried, and spots were counted using a light microscope. PBMCs incubated with DPBS or Phytohemagglutinin (PHA; provided by the manufacturer) were used as negative and positive controls.

### 4.8. Multiplex Analysis

Serum cytokine levels were analyzed using Bio-Plex (Bio-Rad, Hercules, CA, USA) multiplex magnetic bead-based antibody detection kits following the manufacturer’s instructions. Multiplex kit Bio-Plex Pro Human Cytokine 48-plex Screening Panel (12007283, BioRad, Hercules, USA) was used to detect serum cytokines. Serum aliquots (50 μL) were analyzed with a minimum of 50 beads per analyte acquired. Median fluorescence intensities were collected using a MAGPIX analyzer (Luminex, Austin, TX, USA). Each sample was analyzed in triplicate. Data collected was analyzed with MasterPlex CT control software and MasterPlex QT analysis software (MiraiBio, San Bruno, CA, USA). Standard curves for each cytokine were generated using standards provided by the manufacturer.

### 4.9. Mapping of Reactive Peptides 

The three-dimensional structure of SARS-CoV-2 spike glycoprotein (PDBid: 6VXX) was used as the base structure to locate the reactive peptides. All the peptides were mapped on the base structure using PyMOL.

### 4.10. Conservancy and Mutational Analysis of Reactive Peptides

SARS-CoV-2 S protein sequences of recently circulating strains were downloaded from the NCBI virus database. The conservation of each reactive peptide to these circulating strains was analyzed using the IEDB epitope conservation analysis tool. The percent of conservation refers to the changes in a peptide aa sequence compared to the reference sequence (YP_009724390.1) of SARS-CoV-2 S protein. Percentage of conservation was calculated with the following equation:(1)$$\mathrm{Conservation}\left(\%\right)\mathrm{ofpeptide}=\frac{\mathrm{Identical}\;\mathrm{peptide}\;\mathrm{sequence}\;\mathrm{present}\;\mathrm{in}\;\mathrm{number}\;\mathrm{of}\;\mathrm{sestrains}}{\mathrm{total}\;\mathrm{number}\;\mathrm{of}\;\mathrm{strains}}*100$$

Fifty-two circulating strains (Appendix A) were aligned to the reference sequence (YP_009724390.1) using the ViPRbrc database and then visualized in Bioedit 7.2 to locate the mutations. Mutation frequency for each mutated position was calculated by taking the ratio of the number of strains having that mutated position divided by the total number of strains. Each reactive peptide was examined for the presence/absence of these mutations.

### 4.11. Statistical Analysis

Statistical analysis was performed in the R environment [78]. Statistically significant differences between comparison groups were accepted as *p* < 0.05, assessed by the Kruskal–Wallis test with Benjamini–Hochberg adjustment for multiple comparisons. The principal component analysis (PCA) was carried out using the R packages ggplot2 (version 3.3.3) [79] and ggfortify (version 0.4.11) [80].

## Figures and Tables

**Figure 1 ijms-22-11211-f001:**
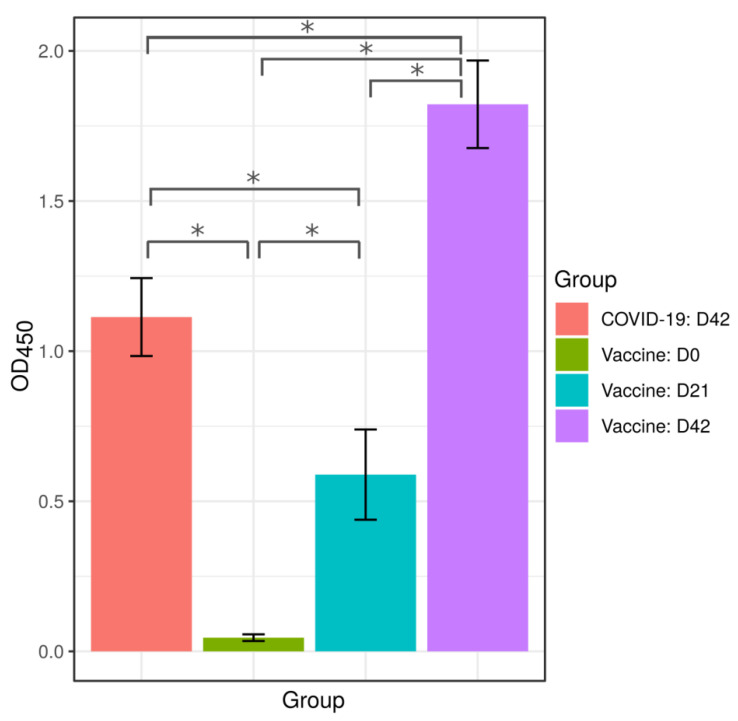
Analysis of anti-SARS-CoV-2 antibodies in sera of convalescent COVID-19 and SputnikV immunized individuals. SputnikV immunized sera (40 samples) were collected on Day 0 (D0; green bar), Day 21 (D21; blue bar) and Day 42 (D42; purple bar) after immunization. Sera from convalescent COVID-19 patients (40 samples; red bar) were collected between 32 and 65 days after the first symptoms (median 42.0 ± 11.1 days). Anti-SARS-CoV-2 IgM, IgG and IgA antibodies were detected using the Coronapass ELISA kit (BioPalitra, NextGene Russia). Sera from the same individuals before immunization was used as control. Data are presented as Mean ± SEM (standard error of the mean). Y-axis shows OD_450_ values (Coronapass ELISA kit; BioPalitra, NextGene Russia * statistical significance).

**Figure 2 ijms-22-11211-f002:**
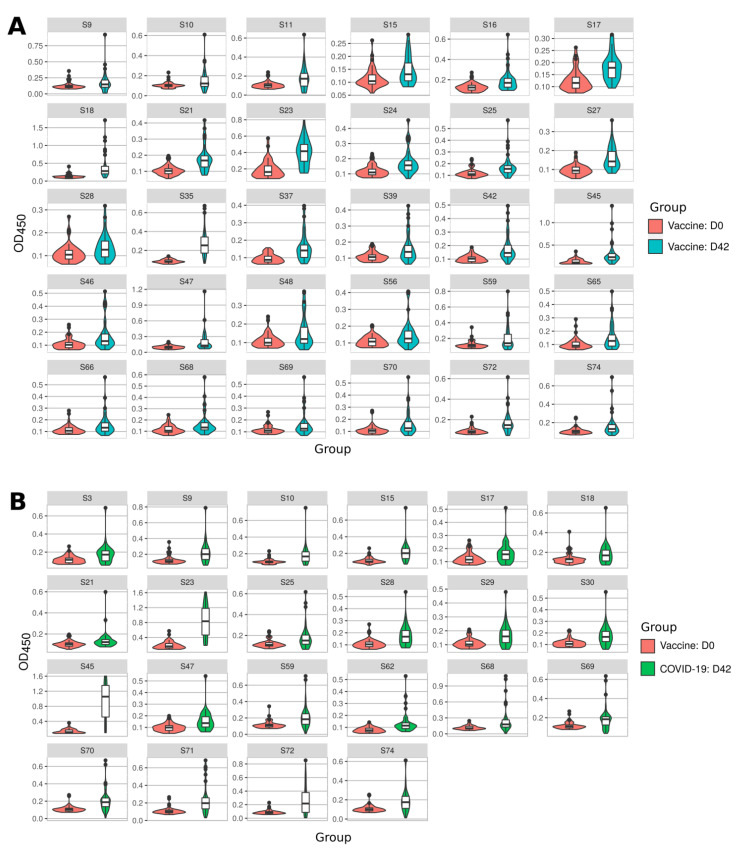
Reactivity to SARS-CoV-2 S protein peptides in SputnikV vaccinated and convalescent sera. Serum samples (40 samples) were collected on D0 and D42 after SputnikV immunization. Serum from convalescent COVID-19 patients (40 samples) was collected between 32 and 65 days after recovery (42.0 ± 11.1 days). Data are presented as Violin plots with boxplots of measured OD650 values. (**A**) SARS-CoV-2 peptide reactivity to sera before (D0) and after (D42) SputnikV vaccination. (**B**) SARS-CoV-2 peptide reactivity to (D0) vaccination and convalescent COVID-19 sera. (**C**) SARS-CoV-2 peptide reactivity between SputnikV vaccinated (D42) and convalescent COVID-19 sera. (**C-I**) reactivity to SARS-CoV-2 peptides was higher in COVID-19 convalescent as compared to vaccinated sera; (**C-II**) reactivity to SARS-CoV-2 peptides was higher in vaccinated as compared to COVID-19 convalescent sera. All presented peptides have statistically significant differences in comparison groups (*p* < 0.05, Kruskal–Wallis test).

**Figure 3 ijms-22-11211-f003:**
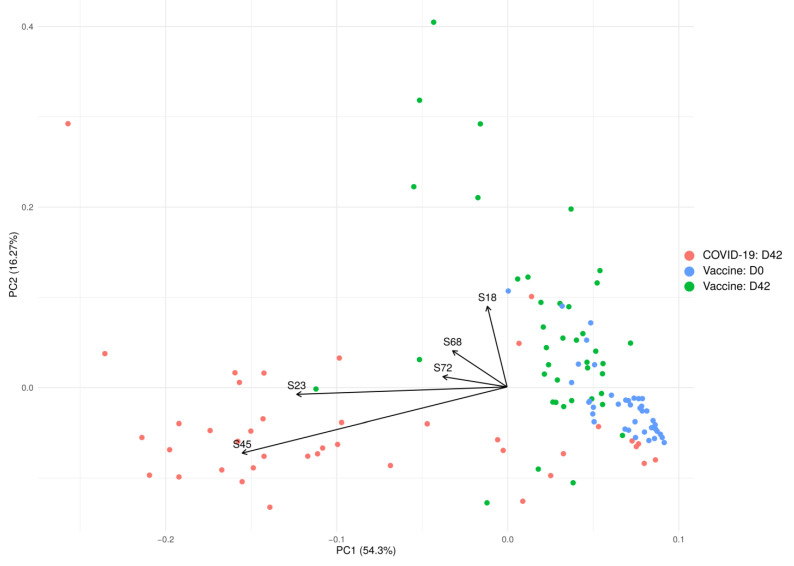
PCA based on the levels of anti-SARS-CoV-2 antibodies in SputnikV vaccinated and convalescent COVID-19 sera. The top five variables (arrows) with the highest contribution are presented. PCA reduces the dimension of multidimensional data (46 peptides) to two (2 axis—PC1 and PC2), which could be visualized with minimal loss information.

**Figure 4 ijms-22-11211-f004:**
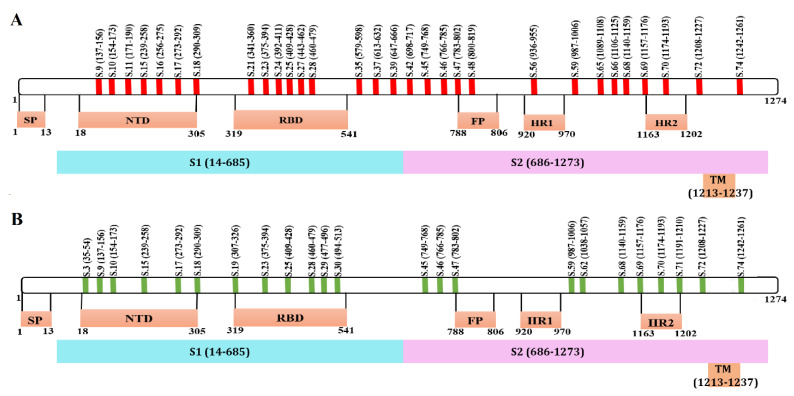
Schematic presentation of SARS-CoV-2 S protein peptides reacting with SputnikV immunized and convalescent COVID-19 sera. Location of peptides recognized by (**A**) SputnikV immunized serum and (**B**) convalescent COVID-19 sera.

**Figure 5 ijms-22-11211-f005:**
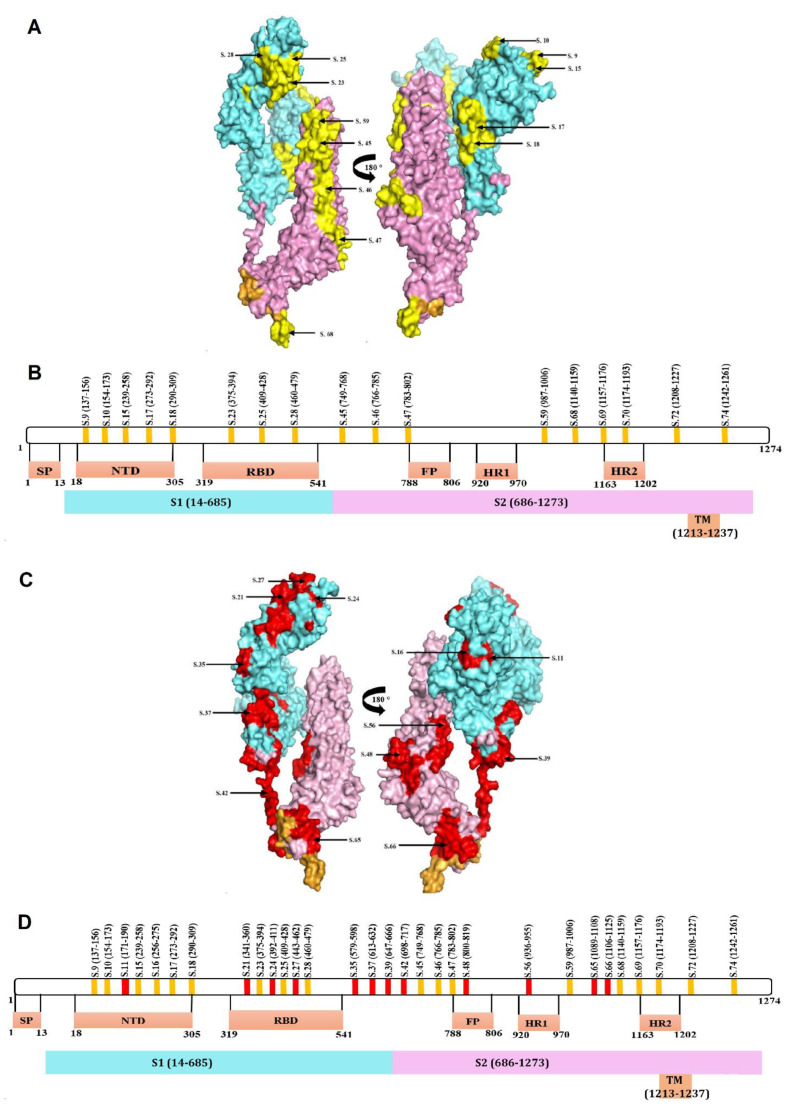
Mapping of peptides recognized by SputnikV immunized and convalescent COVID-19 sera on the three-dimensional SARS-CoV-2 S protein model (PDBid: 6VXX). (**A**) Peptides recognized by both vaccine and convalescent sera (yellow) mapped onto a 3D model of SARS-CoV-2 S protein and (**B**) a linear schematic indicating the location of the peptides in the different domains. (**C**) Peptides recognized only with SputnikV vaccinated sera (red) mapped onto the 3D SARS-CoV-2 S protein model and (**D**) a linear schematic indicating the location of peptides identified in only vaccinated (red) and in both vaccinated and COVID-19 sera (yellow). (**E**) Peptides recognized only with COVID-19 convalescent sera (green) mapped onto the 3D SARS-CoV-2 S protein model and (**F**) a linear schematic indicating the location of peptides identified in only COVID-19 convalescent (green) and both vaccinated and COVID-19 sera (yellow). Blue—S1 domain; Pink—S2 domain; Yellow—TM domain; SP: Signal peptide; NTD: N terminal Domain; RBD: Receptor-binding domain; TM: Transmembrane domain; FP: fusion peptide; HR: heptad repeat.

**Figure 6 ijms-22-11211-f006:**
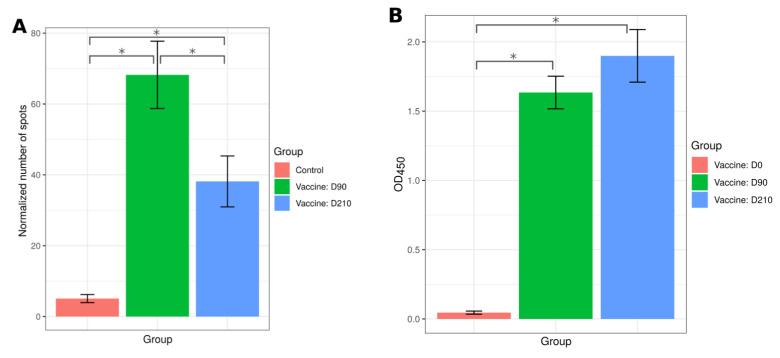
Analysis of the T cell and humoral immune response at day 90 (D90) and day 210 (D210) after SputnikV vaccination. (**A**) T cell immune response. PBMCs from 26 vaccinated (D90 and D210) and 12 SARS-CoV-2 antibody-negative controls were used to analyze T cell immune response using the Tigra test SARS-CoV-2 kit (Generium Corporation, Vladimir region, Russia). Spots were counted, the background (spots in unstimulated PBMCs) was subtracted, and results were presented as a normalized number of spots. (**B**) Forty serum samples were collected at day 0 (D0), day 90 (D90) and day 210 (D210) of SputnikV vaccination. Anti-SARS-CoV-2 IgM, IgG and IgA antibodies were detected using the Coronapass ELISA kit (BioPalitra, NextGene Russia). Data are presented as Mean ± SEM (standard error of the mean). Y-axis shows OD450 value (Coronapass ELISA kit; BioPalitra, NextGene Russia) * statistical significance.

**Figure 7 ijms-22-11211-f007:**
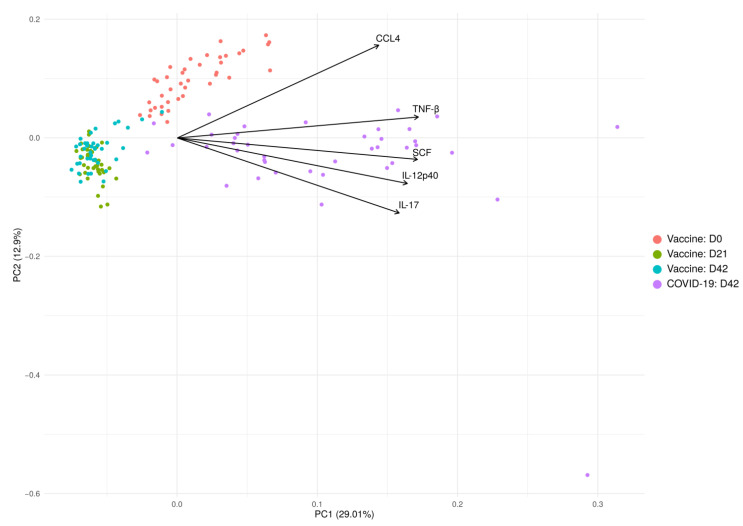
PCA is based on the levels of cytokines in SputnikV vaccinated and convalescent COVID-19 sera. The top 5 variables (arrows) with the highest contribution are presented. PCA reduces the dimension of multidimensional data (48 cytokines) to two (2 axis—PC1 and PC2), which could be visualized with minimal loss information.

**Figure 8 ijms-22-11211-f008:**
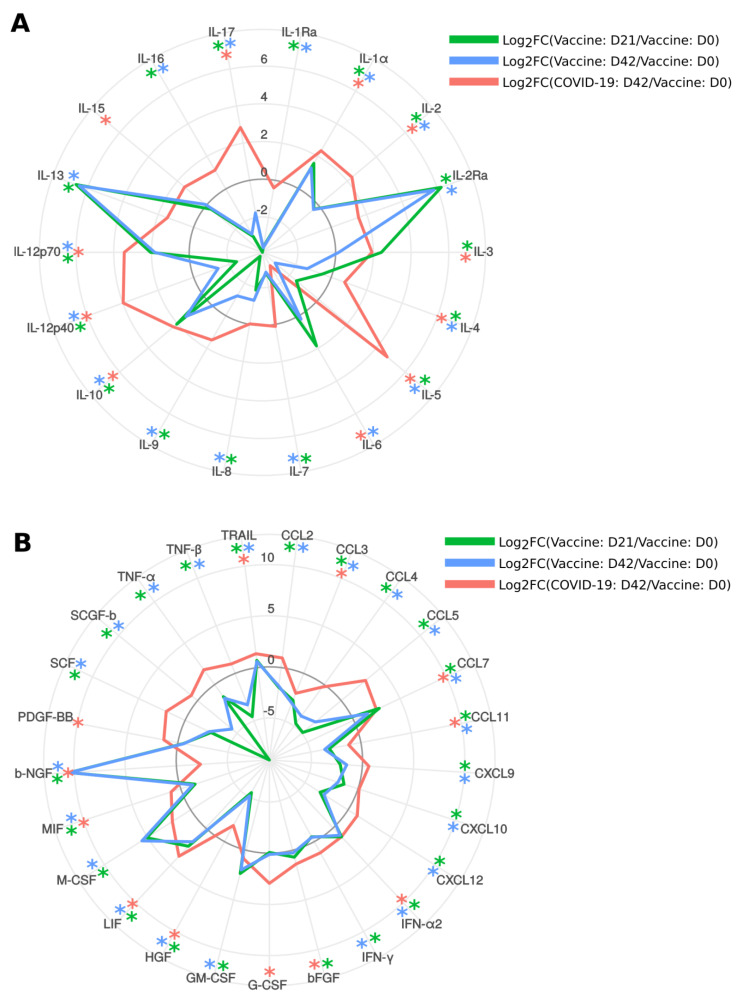
Analysis of cytokines in SputnikV vaccinated and convalescent COVID-19 sera. A and B Serum samples before (D0), after vaccination on D21 and D42 (21 and 42 days after the first dose of vaccine, respectively) as well as 42.0 ± 11.1 days (median ± SEM) after COVID-19 convalescence were used to analyze 48 cytokines (Bio-Plex Pro Human Cytokine 48-plex Screening Panel (12007283, BioRad, Hercules, USA)). Each sample was tested in triplicate. Data were analyzed with MasterPlex CT control software and MasterPlex QT analysis software (MiraiBio, San Bruno, CA, USA). Data are presented as a Log2 difference between a vaccinated or convalescent sample and controls. Cytokines IL-1β, IL-18, CCL27, CXCL1 and VEGF did not differ significantly and were not included in the figure *****—Significant difference between D21 and D0; *****—significant difference between D42 and D0; *****—significant difference between COVID-19 convalescent and D0.

**Figure 9 ijms-22-11211-f009:**
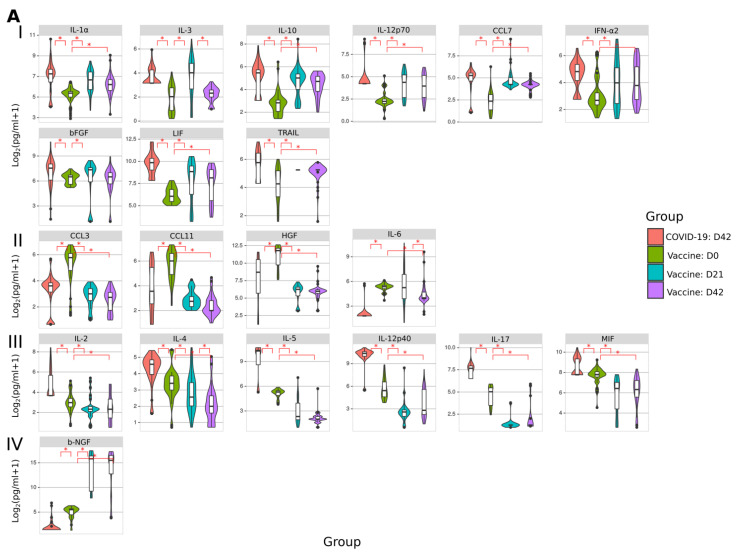
Analysis of cytokines in SputnikV vaccinated and convalescent COVID-19 sera. Serum samples from SputnikV (D21 and D42), as well as 42.0 ± 11.1 days (median ± SEM) after COVID-19 convalescence, were used to analyze 48 cytokines (Bio-Plex Pro Human Cytokine 48-plex Screening Panel). Each sample was tested in triplicate. Data collected was analyzed with MasterPlex CT control software and MasterPlex QT analysis software (MiraiBio, San Bruno, CA, USA). (**A**) Cytokines affected in SputnikV vaccinated and convalescent COVID-19 sera. (**A-I**) cytokines increased after vaccination and convalescent COVID-19; (**A-II**) cytokines decreased after vaccination and convalescent COVID-19; (**A-III**) cytokines decreased after vaccination, while increased in convalescent COVID-19; (**A-IV**) cytokines increased after vaccination, while decreased in convalescent COVID-19. (**B**) Cytokines are affected only after vaccination. **(B, I)** cytokines decreased after vaccination while not changing in convalescent COVID-19; (**B, II**) cytokines increased after vaccination while not changing in convalescent COVID-19. (**C**) cytokines not changed in vaccinated, while increased in convalescent COVID-19. Asterisks (*) indicate statistically significant differences between cytokines (*p* < 0.05, Kruskal–Wallis test).

**Figure 10 ijms-22-11211-f010:**
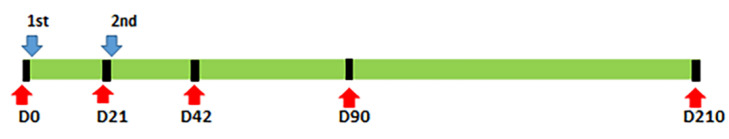
Schematic presentation of vaccination and serum collection.

**Table 1 ijms-22-11211-t001:** Conservation analysis of SARS-CoV-2 S protein peptides recognized by vaccinated and COVID-19 convalescent sera.

Peptides (*n*)	Peptide Name	Strains with 100% Identity (*n*)	Conservation (%) when 52 Strains Were Included
12	S16, S17, S18, S19, S35, S47, S48, S59, S62, S65, S68, S72	52	100.0
4	S45, S69, S70, S71	51	98.1
6	S11, S21, S23, S24, S46, S74	50	96.2
2	S25, S66	49	94.20
2	S39, S56	48	92.30
1	S3	47	90.4
1	S.15	44	84.6
5	S10, S28, S30, S42	43	82.7
1	S.9	41	78.8
1	S27	39	75.00
1	S29	32	61.50
1	S37	3	5.80

*n*—number of peptides.

**Table 2 ijms-22-11211-t002:** Conservation analysis of S protein peptides grouped by type of sera.

	All Reactive Peptides		Vaccinated and COVID-19		COVID-19		Vaccinated	
Groups	Conservation (%)	Peptides (*n*)	Peptides	*n*	Peptides	*n*	Peptides	*n*
1	100	12	S17, S18, S47, S59, S68, S72	6	S19, S62	2	S16, S35, S48, S65	4
2	>90 < 100	15	S23, S25, S45, S46, S69, S70, S74	7	S3, S71	2	S11, S21, S24, S39, S56, S66	6
3	>80 < 90	5	S10, S15, S28	3	S30	1	S42	1
4	>70 < 80	2	S9	1		0	S27	1
5	>60 < 70	1		0	S29	1		0
6	<60	1		0		0	S37	1

*n*—number of peptides.

**Table 3 ijms-22-11211-t003:** Mutation analysis in SARS-CoV-2 S protein peptides recognized by vaccinated and COVID-19 convalescent sera.

Vaccinated	No. of Mutated Positions	Convalescent	No. of Mutated Positions	Vaccinated and Convalescent	No. of Mutated Positions
S11	2	S3	2	S9	7
S21	1	S29	5	S10	3
S24	2	S30	2	S15	2
S27	2	S71	1	S23	2
S37	4			S25	2
S39	3			S28	2
S42	2			S45	1
S56	1			S46	2
S66	2			S69	1
				S70	1
				S74	2

**Table 4 ijms-22-11211-t004:** SputnikV vaccinated and controls demographics information.

Clinical Characteristics	Values
Vaccinated age (years)	43.3 ± 16.4
Vaccinated sex (M/F)	17/23
Convalescent COVID-19 age (years)	39.1 ± 13.2
Convalescent COVID-19 age (M/F)	15/25
Control age (years)	47.1 ± 13.8
Control sex (M/F)	5/7

**Table 5 ijms-22-11211-t005:** Sequence and position of SARS-CoV-2 S protein peptides.

Peptide	Aa Sequence	Position	Peptide	Aa Sequence	Position	Peptide	Aa Sequence	Position
S1	MFVFLVLLPLVSSQCVNLTT	1–20	S25	QIAPGQTGKIADYNYKLPDD	409–428	S50	IKQYGDCLGDIAARDLICAQ	834–853
S2	LTTRTQLPPAYTNSFTRGVY	18–37	S26	PDDFTGCVIAWNSNNLDSKV	426–445	S51	CAQKFNGLTVLPPLLTDEMI	851–870
S3	GVYYPDKVFRSSVLHSTQDL	35–54	S27	SKVGGNYNYLYRLFRKSNLK	443–462	S52	EMIAQYTSALLAGTITSGWT	868–887
S4	QDLFLPFFSNVTWFHAIHVS	52–71	S28	NLKPFERDISTEIYQAGSTP	460–479	S53	GWTFGAGAALQIPFAMQMAY	885–904
S5	HVSGTNGTKRFDNPVLPFND	69–88	S29	STPCNGVEGFNCYFPLQSYG	477–496	S54	MAYRFNGIGVTQNVLYENQK	902–921
S6	FNDGVYFASTEKSNIIRGWI	86–105	S30	SYGFQPTNGVGYQPYRVVVL	494–513	S55	NQKLIANQFNSAIGKIQDSL	919–938
S8	VNNATNVVIKVCEFQFCNDP	120–139	S31	VVLSFELLHAPATVCGPKKS	511–530	S56	DSLSSTASALGKLQDVVNQN	936–955
S9	NDPFLGVYYHKNNKSWMESE	137–156	S32	KKSTNLVKNKCVNFNFNGLT	528–547	S57	NQNAQALNTLVKQLSSNFGA	953–972
S10	ESEFRVYSSANNCTFEYVSQ	154–173	S33	GLTGTGVLTESNKKFLPFQQ	545–564	S58	FGAISSVLNDILSRLDKVEA	970–989
S11	VSQPFLMDLEGKQGNFKNLR	171–190	S34	FQQFGRDIADTTDAVRDPQT	562–581	S59	VEAEVQIDRLITGRLQSLQT	987–1006
S12	NLREFVFKNIDGYFKIYSKH	188–207	S35	PQTLEILDITPCSFGGVSVI	579–598	S60	LQTYVTQQLIRAAEIRASAN	1004–1023
S13	SKHTPINLVRDLPQGFSALE	205–224	S36	SVITPGTNTSNQVAVLYQDV	596–615	S61	SANLAATKMSECVLGQSKRV	1021–1040
S14	ALEPLVDLPIGINITRFQTL	222–241	S37	QDVNCTEVPVAIHADQLTPT	613–632	S62	KRVDFCGKGYHLMSFPQSAP	1038–1057
S15	QTLLALHRSYLTPGDSSSGW	239–258	S38	TPTWRVYSTGSNVFQTRAGC	630–649	S63	SAPHGVVFLHVTYVPAQEKN	1055–1074
S16	SGWTAGAAAYYVGYLQPRTF	256–275	S39	AGCLIGAEHVNNSYECDIPI	647–666	S64	EKNFTTAPAICHDGKAHFPR	1072–1091
S17	RTFLLKYNENGTITDAVDCA	273–292	S41	PRRARSVASQSIIAYTMSLG	681–700	S65	FPREGVFVSNGTHWFVTQRN	1089–1108
S18	DCALDPLSETKCTLKSFTVE	290–309	S42	SLGAENSVAYSNNSIAIPTN	698–717	S66	QRNFYEPQIITTDNTFVSGN	1106–1125
S19	TVEKGIYQTSNFRVQPTESI	307–326	S43	PTNFTISVTTEILPVSMTKT	715–734	S68	PLQPELDSFKEELDKYFKNH	1140–1159
S20	ESIVRFPNITNLCPFGEVFN	324–343	S45	CSNLLLQYGSFCTQLNRALT	749–768	S69	KNHTSPDVDLGDISGINASV	1157–1176
S21	VFNATRFASVYAWNRKRISN	341–360	S46	ALTGIAVEQDKNTQEVFAQV	766–785	S70	ASVVNIQKEIDRLNEVAKNL	1174–1193
S22	ISNCVADYSVLYNSASFSTF	358–377	S47	AQVKQIYKTPPIKDFGGFNF	783–802	S71	KNLNESLIDLQELGKYEQYI	1191–1210
S23	STFKCYGVSPTKLNDLCFTN	375–394	S48	FNFSQILPDPSKPSKRSFIE	800–819	S72	QYIKWPWYIWLGFIAGLIAI	1208–1227
S24	FTNVYADSFVIRGDEVRQIA	392–411	S49	FIEDLLFNKVTLADAGFIKQ	817–836	S74	SCLKGCCSCGSCCKFDEDDS	1242–1261
						S75	CKFDEDDSEPVLKGVKLHYT	1254–1273

## Data Availability

The study did not report any data.

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
