# Peer review of "Long Term Immune Response Produced by the SputnikV Vaccine"

_ijms, 2021, doi:10.3390/ijms222011211_

Round 1

Reviewer 1 Report

The sputnik vaccine is not the only adenovirus based vaccine against SARS-COV2, nor the first. The authors should cite studies by others (e.g. Ewer, K.J., et al., T cell and antibody responses induced by a single dose of ChAdOx1 nCoV-19 (AZD1222) vaccine in a phase 1/2 clinical trial. Nature medicine, 2021. 27(2): p. 270-278.)

Figure 1 and Table S1. Why is the result calculated as a ratio? The authors state this is the manufacturer’s instruction. However, it inflates the apparent readings, which would be more usefully displayed as simple OD450, with a horizontal line indicating the mean negative control value.  Is the statistical analysis performed using the calculated ratios, or the absorbance data? It should be performed using the absorbance data, possibly with the negative control mean subtracted.

Figure 6B. As for figure 1.

Lines 467 to 471. The vaccine recipients are exposed to multiple viral antigens from the adenovirus vector.  So the difference could also be due to qualitative differences between Adenovirus and SARS-COV2, or to the quantity of coronavirus antigens encountered in an active covid infection, as well as the absence of other SARS-COV2 antigens. This should be reflected in the text.

Line 68  “…epitopes of SARS-CoV-2 S protein elicts an antibody…” change to “…epitopes of SARS-CoV-2 S protein elict an antibody…”

Line 276 “…revealed that changes in serum cytokines is similar in vaccinated…” change to “…revealed that changes in serum cytokines are similar in vaccinated…”

Lines 362-363  font size variation

Lines 365  font size variation

Line 366  State what virus the referenced studies were investigating

Line 368 “…contain antigens eliciting…” change to “…contain epitopes eliciting…”

Lines 385-392  font size variation

Lines 388 – 471.  Needs checking for grammar (e.g. Line 464-465 “…It should be noted that convalescent COVID-19 had more cytokines upregulated…”   should be “…It should be noted that convalescent COVID-19 sera had more cytokines upregulated…”

Author Response

Point by point reply for ijms-1382749 Martynova et al

Reviewer 1:

Q1: The sputnik vaccine is not the only adenovirus based vaccine against SARS-COV2, nor the first. The authors should cite studies by others (e.g. Ewer, K.J., et al., T cell and antibody responses induced by a single dose of ChAdOx1 nCoV-19 (AZD1222) vaccine in a phase 1/2 clinical trial. Nature medicine, 2021. 27(2): p. 270-278.)

Author’s response. We have added a line to the introduction stating that other types of vaccines have also been developed (Line 60-63)

Q2: Figure 1 and Table S1. Why is the result calculated as a ratio? The authors state this is the manufacturer’s instruction. However, it inflates the apparent readings, which would be more usefully displayed as simple OD450, with a horizontal line indicating the mean negative control value.  Is the statistical analysis performed using the calculated ratios, or the absorbance data? It should be performed using the absorbance data, possibly with the negative control mean subtracted.

Author’s response. The reviewer is correct that the ratio calculation is recommended by the manufacturer. Whilst the numbers previously presented were larger, the statistical differences are the same regardless of whether OD or ratio is used. For transparency we have changed the figure to OD values.

Q3: Figure 6B. As for figure 1.

Author’s response. For continuity we have changed this figure to OD values as well. As for Figure 1 interpretation of the data has not changed.

Q4: Lines 467 to 471. The vaccine recipients are exposed to multiple viral antigens from the adenovirus vector. So the difference could also be due to qualitative differences between Adenovirus and SARS-COV2, or to the quantity of coronavirus antigens encountered in an active covid infection, as well as the absence of other SARS-COV2 antigens. This should be reflected in the text.

Author’s response. We agree that this is a valid point worth making and have added the following statement to lines 538-544 “Additionally, differences in cytokine activation could be explained by the differing quantities of SARS-CoV-2 antigens encountered during infection and upon vaccination. Some findings support this suggestion where a higher viral load was closely related to severe COVID-19 (Liu et al., 2021; Zheng et al., 2020). Also, the simultaneous exposure to adenovirus antigens from the vector and SARS-CoV-2 antigens could affect cytokine activation in vaccinated individuals.”

Q5: Line 68  “…epitopes of SARS-CoV-2 S protein elicts an antibody…” change to “…epitopes of SARS-CoV-2 S protein elict an antibody…”

Q6: Line 276 “…revealed that changes in serum cytokines is similar in vaccinated…” change to “…revealed that changes in serum cytokines are similar in vaccinated…”

Q7: Lines 362-363  font size variation

Q8: Lines 365  font size variation

Q9: Line 366  State what virus the referenced studies were investigating

Q10: Line 368 “…contain antigens eliciting…” change to “…contain epitopes eliciting…”

Q11: Lines 385-392  font size variation

Q12: Lines 388 – 471.  Needs checking for grammar (e.g. Line 464-465 “…It should be noted that convalescent COVID-19 had more cytokines upregulated…”   should be “…It should be noted that convalescent COVID-19 sera had more cytokines upregulated…”

Author’s response. We thank the reviewer for careful reading of the manuscript and highlighting these typographical errors. We have corrected the errors noted, carefully re-read the manuscript, and changed the grammatical errors. We apologise for any irritation caused by the formatting errors created by the upload and formatting tool of the website. The authors had not realised they had appeared.

Reviewer 2 Report

Martynova et al in the paper titled "Long term immune response produced by the SputnikV vaccine" enrolled 40 individuals to analyze the vaccine efficacy of vaccine SputnikV in terms of T cell response and humoral response. I have the following points

Minor:

  1. The format of the papers appears really a rough draft where some paragraphs are in a different format than others. This reflects poorly on the editing team
  2. A number of citations are general in nature and original authors are not represented in the citation report
  3.  The figures are poorly represented and the resolution is not quite high in some of the figures.                                                                                   Major: 1. The sample size is too small and the authors did not classify subject and their demographies accordingly. Maybe a table will help2. Authors started with 40 subjects, but the data represented in fig. is not from all the subjects. What happened to other subjects?
    3. Fig one missing axis and can use better legend explanation. Why not initially expand the "D" (e. D0 to Day 0) to increase readability?
  4. What is the real-world data post-vaccination on follow-up studies? How long does the immunity last? Did cytokines generate any complications?
  5. Finally, in a lot of places, the formatting reflects as if it is copy-pasted from one place to another. There are comma and periods missing throughout the manuscript and reflect a poor job on the author's part.

Author Response

Point by point reply for ijms-1382749 Martynova et al

Reviewer 2:

Martynova et al in the paper titled "Long term immune response produced by the SputnikV vaccine" enrolled 40 individuals to analyze the vaccine efficacy of vaccine SputnikV in terms of T cell response and humoral response. I have the following points

Minor Q1: The format of the papers appears really a rough draft where some paragraphs are in a different format than others. This reflects poorly on the editing team

Author’s response. We thank the reviewer for careful reading of the manuscript and highlighting these formatting errors. We apologise for any irritation caused by the formatting errors created by the upload and formatting tool of the website. The authors had not realised they had appeared.

Minor Q2: A number of citations are general in nature and original authors are not represented in the citation report

Author’s response. The references have been checked and replaced with primary sources where appropriate

Minor Q3: The figures are poorly represented and the resolution is not quite high in some of the figures.

Author’s response. The quality of all figures was evaluated and modified according to the MDPI standards. In general, the quality of figures could be affected when they are embedded into the Word file. We have submitted individual figures, to make sure that the quality of the original figures follows the Journal’s requirements.

Major Q1:  The sample size is too small and the authors did not classify subject and their demographies accordingly. Maybe a table will help

Author’s response. The authors acknowledge the sample size is not huge but feel that the statistical findings drawn from the data are robust. We agree that a table of demographic information would be useful, and this has been added as Table 4. We feel it is worth noting that the authors started to collect samples as soon as the SputnikV vaccine was available in Kazan, Republic of Tatarstan. Collection of samples was restricted as we were only allowed to collect at single designated site. At the beginning of vaccination campaign, there were very few individuals willing to be vaccinated and even less wanted to participate in this study. We believe that publishing this data, even with restricted number of participants, is very important because:

  1. It provides health care workers with information about safety and efficacy of the vaccine, which is important to deliver science-based information to the public
  2. It provides the general population with valid scientific information that will help them to make an informed decision on vaccination
  3. It is important for the scientific community as it provides information from the ‘real life’ administration of the vaccine
  4. Importantly, there is limited information on SputnikV vaccine efficacy, which is a major gap in our understanding of this vaccine potential to control the SARS-CoV-2 epidemic. Altogether, making data on SputnikV vaccine public is an important contribution to combat the SARS-CoV-2 pandemic, especially in Russia

Major Q2. Authors started with 40 subjects, but the data represented in fig. is not from all the subjects. What happened to other subjects?

Author’s response. All 40 subjects were used for all analysis except for Figure 6. We apologise for not making this clearer in the text and figure legend. This has now been added. Unfortunately, participation levels decreased so the authors were limited on numbers. However, we have extended the data to now include T cell and humoral analysis out to 210 days post first dose of vaccination, which shows that the level of sustained immunity is still measurable at this later timepoint.

Major Q3: Fig one missing axis and can use better legend explanation. Why not initially expand the "D" (e. D0 to Day 0) to increase readability?

Author’s response. We thank the reviewer for the careful reading. The x-axis now has the label “Group”. The figure legend was modified to improve the readability. We added a sentence to explain how samples were collected in lines 92-95: “Samples used for this analysis were collected on days 21 and 42 after the first vaccine dose and labelled as D21 and D42, respectively. The day of the first vaccine dose was indicated as day 0 and marked as D0.”

Major Q4: What is the real-world data post-vaccination on follow-up studies? How long does the immunity last? Did cytokines generate any complications?

Author’s response. For the Sputnik V vaccine there is a dearth of information on ‘real-world’ data, hence the value in this manuscript. During the time that this manuscript has undergone review, we have obtained further samples from 26 participants 210 days after vaccination and analysed the humoral and T cell response. These data have been added to Figure 6 and provides insight into the extremely relevant point raised by the reviewer “how long does immunity last”. We can see that both T cell and antibody response still exist at this time point. Further research is warranted to give information on longer time since vaccination but is outside the scope of this study. The vaccinated individuals did not describe any health issues related to the vaccination at any point and did not have measurable changes in their serum cytokine levels at D210.

Major Q5: Finally, in a lot of places, the formatting reflects as if it is copy-pasted from one place to another. There are comma and periods missing throughout the manuscript and reflect a poor job on the author's part.

Author’s response. We thank the reviewer for careful reading of the manuscript and highlighting these typographical errors. We have corrected the errors noted, carefully re-read the manuscript, and changed the grammatical errors. We apologise for any irritation caused by the formatting errors created by the upload and formatting tool of the website. The authors had not realised they had appeared.

Round 2

Reviewer 2 Report

More real-world data can be added given the distribution of this vaccine to multiple countries. Why focus only on a small region?

Author Response

Reviewer 2:

Query: More real-world data can be added given the distribution of this vaccine to multiple countries. Why focus only on a small region?

Author’s answer. There are limited published data on the real life efficacy of SputnikV vaccine. Still, we did the published data analysis in order to address Reviewer’s comments. This analysis was added into the Discussion section (lines 520-535.) “There is limited real-life data on the duration of the immune response elicited by the Sputnik V vaccine. Our data, therefore, collected on samples from 90 and 210 days after the first dose of SputnikV vaccine demonstrating long term activation of both the humoral and T cell immune responses is of importance. These long-term immune responses could explain the efficacy of the SputnikV vaccine reported by Gonzalez et al in an Argentinian cohort (González, S., et al.2021). Authors have also shown that receiving even a single dose of vaccine reduced duration of hospitalization and fatality, after exposure to SARS-CoV-2. In another study by Rossi et al, eliciting anti-SARS-CoV-2 neutralizing antibodies was demonstrated in 288 volunteers 21 days after the first dose vaccine (Rossi, A. H., et al. 2021). Also, the efficacy of SputnikV was demonstrated in cohort from Venezuela, where even a single dose was shown sufficient to induce neutralizing antibodies in previously SARS-CoV-2 positive individuals (Claro, F., et al. 2021). A 100% seroconversion was found in a Venezuelan cohort 6 weeks after the second dose of SputnikV vaccine, confirming induction of long term of immune response.